

# A randomized trial to evaluate the impact of copra meal hydrolysate on gastrointestinal symptoms and gut microbiome

Witida Sathitkowitchai[1,2,3], Narissara Suratannon[4], Suttipun Keawsompong[1,2], Wanlapa Weerapakorn[4], Preecha Patumcharoenpol[5], Sunee Nitisinprasert[1,2] and Massalin Nakphaichit[1,2]

[1] Department of Biotechnology, Faculty of Agro-Industry, Kasetsart University, Bangkok, Thailand
[2] Center for Advanced Studies for Agriculture and Food, Kasetsart University Institute for Advanced Studies, Kasetsart University, Bangkok, Thailand
[3] Microarray Research Team, National Center for Genetic Engineering and Biotechnology, Thailand Science Park, Pathum Thani, Thailand
[4] Pediatric Allergy & Clinical Immunology Research Unit, Division of Allergy and Immunology, Department of Pediatrics, Faculty of Medicine, Chulalongkorn University, King Chulalongkorn Memorial Hospital, The Thai Red Cross Society, Bangkok, Thailand
[5] Interdisciplinary Graduate Program in Bioscience, Faculty of Science, Kasetsart University, Bangkok, Thailand

Corresponding author
Massalin Nakphaichit,
fagimln@ku.ac.th

## ABSTRACT

The impact of copra meal hydrolysate (CMH) on gut health was assessed by conducting a double-blinded, placebo-controlled study. Sixty healthy adult participants, aged 18–40 years were assigned to daily consume 3 g of CMH, 5 g of CMH or placebo in the form of drink powder for 21 days. Consumption of CMH at 3 g/d improved defecating conditions by reducing stool size and also relieved flatulence and bloating symptoms. Fecal samples were collected serially at the baseline before treatment, after the treatment and after a 2-week washout period. The gut microbiomes were similar among the treatment groups, with microbial community changes observed within the groups. Intake of CMH at 3 g/d led to increase microbial diversity and richness. Reduction of the ratio between *Firmicutes* to *Bacteroidetes* was observed, although it was not significantly different between the groups. The 3 g/d CMH treatment increased beneficial microbes in the group of fiber-degrading bacteria, especially human colonic *Bacteroidetes*, while induction of *Bifidobacteriaceae* was observed after the washout period. Intake of CMH led to increase lactic acid production, while 3 g/d supplement promoted the present of immunoglobulin A (IgA) in stool samples. The 3 g daily dose of CMH led to the potentially beneficial effects on gut health for healthy individuals.

## INTRODUCTION

The human intestine contains trillions of microbes that have been linked to many aspects of health and disease, including metabolic syndrome, infectious diseases and

noncommunicable diseases (*Jarett et al., 2019*). Prebiotics are considered to be functional foods that can modulate the gut microbiome (*Prayoonthien et al., 2019*). The definition of prebiotics by The International Scientific Association of Probiotics and Prebiotics (ISAPP) is "a substrate that is selectively utilized by host microorganisms conferring a health benefit" (*Gibson et al., 2017*). Prebiotics increase levels of beneficial bacteria and positively impact the composition of resident microbiota, while also promoting microbe-derived metabolites or signal molecules such as short chain fatty acids (SCFAs) that are important for gut health (*Gibson et al., 2017*; *Markowiak-Kopeć & Ślizewska, 2020*; *Topping, 1996*). Prebiotics have a mild laxative effect on bowel habits due to indigestible properties. Some prebiotics, especially inulin and oligofructose, alleviate constipation by increasing the fecal bulk (*Andersson et al., 2001*; *Cummings & Macfarlane, 2002*).

Copra meal contains galactomannan as a byproduct from the coconut milk industry. To improve the functional food properties of copra meal, galactomannan was successfully hydrolyzed to mannooligosaccharides (MOS) by the enzyme β-mannanase (*Pangsri et al., 2015*; *Rungruangsaphakun & Keawsompong, 2018*; *Titapoka et al., 2007*). Copra meal hydrolysate (CMH) has been hypothesized as a potential prebiotic. CMH was stable under upper human gastrointestinal tract conditions (*Prayoonthien et al., 2019*). *In vitro* human fecal fermentation showed that CMH promoted the growth of *Lactobacilli* and *Bifidobacteria* similar to fructooligosaccharides (FOS). Induction of two beneficial microbes enhanced SCFAs production (*Prayoonthien et al., 2019*). CMH also suppressed pathogenic bacteria such as *Salmonella*, *Escherichia coli*, *Staphylococcus aureus* and *Shigella dysenteriae* (*Prayoonthien et al., 2019*).

Comparing with other prebiotics, documentation of the health-promoting effect of MOS on humans is limited. Impact of MOS from coffee hydrolysate revealed that MOS supplementation over for three weeks promoted the growth of *Lactobacilli* and *Bifidobacteria* (*Walton et al., 2010*). MOS derived from coffee mannan reduced fat absorption in humans (*Asano et al., 2006*), while MOS reduced the concentration of triglycerides in serum, and subsequently inhibited absorption of lipids in subjects fed a high-fat diet (*Kumao et al., 2006*).

No official dose recommendation of prebiotics in human diet has been reported. Most prebiotics require an oral dose of 3–5 g/d to confer a gut health benefit, depending on prebiotic structure. Daily dosages are suggested at around 5 g for fructooligosaccharides (FOS), galactooligosaccharides (GOS) and plant sources of prebiotics (*Gibson et al., 2017*). This is the first clinical study to evaluate the impact of daily consumption of CMH over three weeks on the gut microbiome, defecating conditions, and gastrointestinal symptoms in healthy volunteers. Different doses of daily consumption were also investigated.

## MATERIALS AND METHODS

### Preparation of copra meal hydrolysate and placebo drinks

Copra meal hydrolysate (CMH) was prepared following *Rungruangsaphakun & Keawsompong (2018)*. The CMH powder contained 9.7% crude protein, 1% crude fat, 14.5% crude fiber, 12.3% ash and 53% nitrogen-free extract. The nitrogen-free extract

was composed of 3-6 molecules of MOS, mainly mannose, linking together. Amount of total oligosaccharide of CMH was 14.77%; mainly mannooligosaccharide with 3-6 degrees of polymerization. The CMH drinks were prepared in three formulas: the placebo group (maltodextrin 10 g), the 3 g CMH group (3 g CMH and 7 g maltodextrin) and the 5 g CMH group (5 g CMH and 5 g maltodextrin). The products were packaged in individual daily-use sachets of identical appearance and completely dissolved in room temperature drinking water before consumption.

## Study design

This study was a randomized double-blinded placebo-controlled trial with a three-arm parallel group design in 1:1:1 ratio as the placebo group, CMH 3 g (3CMH group) and CMH 5 g (5CMH group) (Fig. 1). Healthy adults, aged between 18 and 40 years with a body mass index (BMI) of 18.5 to 25.0 kg/m$^2$, were enrolled (Table S1). This study excluded those who (1) received any antibiotics or anti-viral agents within 3 months before the study, (2) received probiotics/prebiotics within 1 month before the study, and (3) were allergic to coconut and soybean which were components of the study product. Recruitment was conducted through poster and web-based advertising. The study was carried out at the outpatient clinic of King Chulalongkorn Memorial Hospital, Bangkok, Thailand. Physicians and study nurse who involved in the study explained the protocol and enrolled participants to the study. Written informed consent was obtained from all participants. The study was approved by the Ethics Committee of King Chulalongkorn Memorial Hospital, Bangkok, Thailand, under approval reference number 388/61 and registered on the Thai Clinical Trials Registry (TCTR20190426003).

## Sample size calculation

The sample size was determined using the tests for two independent means (two-tailed test) as formula described below. On the basis of previously published study of the effect of MOS from coffee mannan on fecal microflora (*Asano et al., 2006*), the mean change of stool *Bifidobacterium* from baseline of subjects received MOS 3 g ($\mu 1$) was 20%, while for those received MOS 1g ($\mu 2$) was 10%. The difference ($\Delta$) was used to calculate, with a power of approximately 80% (beta 0.2) and a significance level of alpha 0.05. The ratio (r) was 1. SD was estimated to be 8. The numbers of study participants to be included was 11 subjects per group. With the anticipation of 20% drop out rate, at least 14 subjects per group should be included. So, in our study, we included 20 subjects per group.

$$n_1 = \frac{(z_{1-\frac{\alpha}{2}} + z_{1-\beta})^2 [\sigma_1^2 + \frac{\sigma_2^2}{r}]}{\Delta^2}$$

$$r = \frac{n_2}{n_1}, \Delta = \mu_1 - \mu_2$$

## Randomization and allocation concealment

Each eligible participant was randomized to one of three groups and assigned a unique study number. A blocked randomization list was created using Sealed Envelope Ltd.
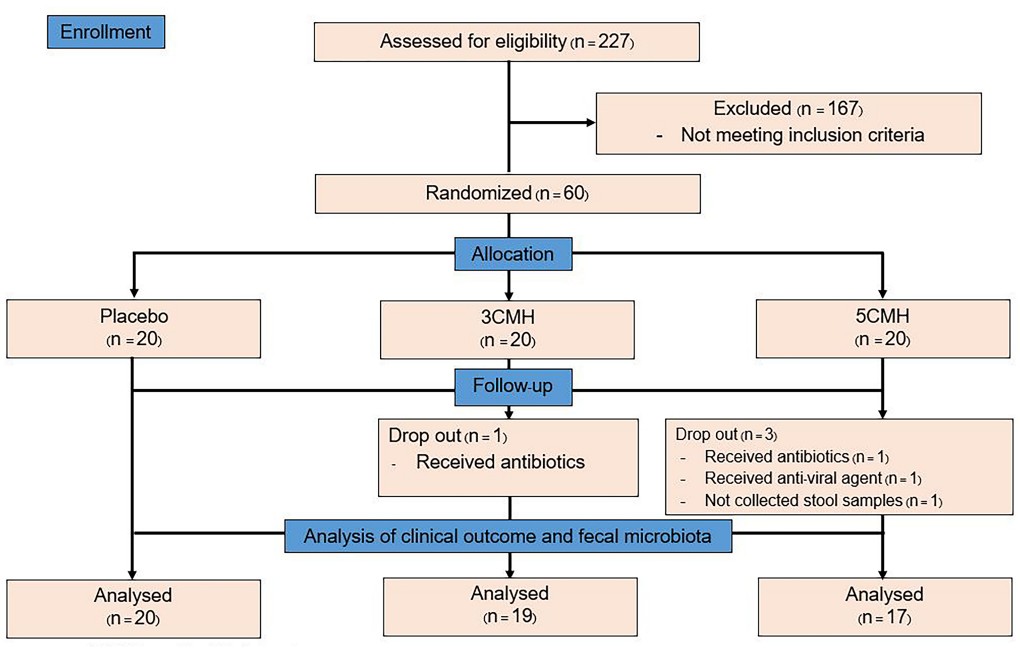

**Figure 1** Flowchart design of the copra meal hydrolysate (CMH) study.

(https://www.sealedenvelope.com), with block sizes of 6 by the statistician who was not involved in the study. Study nurse who did not aware of the code group assigned participants to interventions.

## Intervention

After randomization, participants were subjected to a 2-week baseline period, 3-week treatment period and 2-week washout period with instruction to use the products once daily during the treatment period (Fig. 2). Participants were asked to fill out a daily stool record chart and a 24-hour diary record 3 times per week (2 weekdays and 1 day during weekend) throughout the study period. Dietary restrictions were imposed throughout the entire study period (total 49 days), with prohibition of consumption of other fermented products, probiotics, prebiotics and synbiotics supplements to limit potential interference with evaluation of the testing product. The participants were excluded from the study if (1) received dietary supplements described above during the study, (2) received any antibiotics or anti-viral agents during the study and (3) failed to collect stool samples according to the study protocol. Each participant visited the clinic weekly throughout the study. Stool samples were collected at the baseline visit, the end of treatment, and washout periods. Subjects and researchers who had access to the outcomes of the study were blinded to the code group assigned to participants.

## Stool record chart

The stool record charts, adapted from *Walton et al. (2010)*, were recorded daily by all participants throughout the study, in order to investigate whether the products affected

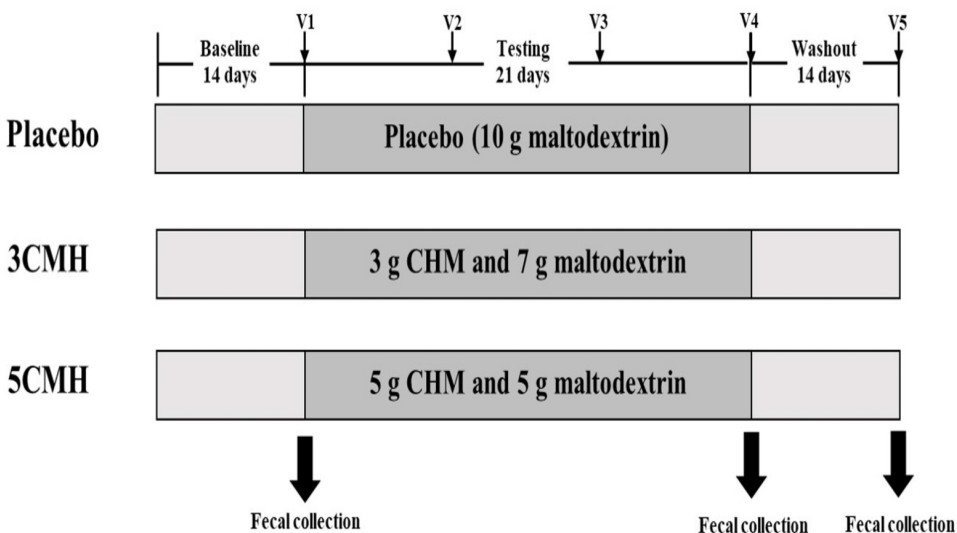

**Figure 2** **Timeline of copra meal hydrolysate (CMH) studied, including sample groups and fecal collection.**

the stool frequency, diarrhea, constipation, and gastrointestinal discomfort (for examples bloating symptoms) of the subjects (Table S2). Each stool characteristic was scored as a number and calculated as amean (±SD) per week.

## 24-hour dietary record

The 24-hour dietary records were filled in by the subjects three times per week (2 weekdays and 1 day during weekend) throughout the study period. Energy and nutritive values were calculated using the nutrient calculation computer software INMUCAL-Nutrients V4.0 database NB. 4 (Food Composition Database for INMUCAL Program, 2018) (Table S3).

## Stool collection and processing

Fresh stools were collected at the baseline, at the end of the treatment period and the end of the washout period for analysis of microbiota, SCFAs and secretory immunoglobulin A (IgA). A stool sample of 20 g was collected in a 76× 20 mm sterile container and immediately placed on ice for transfer to storage at −80 °C. Stool samples were prepared following the method of *Kisuse et al. (2018)* for microbiota and SCFA analyses. The fecal samples were diluted 10-fold with phosphate-buffered saline (pH 8.0) using a stomacher blender (Stomacher® 80 Biomaster; Seward, Worthing, UK) for 5 min. A 1 ml sample of fecal slurry was then centrifuged at 13,000×g for 5 min., and the supernatant was stored at −80 °C for subsequent SCFA analysis. For microbial DNA extraction, 1 ml of fecal slurry was placed in a 1.5 ml centrifuge tube and then stored at −80 °C.

## Microbial DNA extraction and 16S rRNA gene sequencing

Microbial DNA was extracted according to the modified method of *Mirsepasi et al. (2014)*, utilizing a bead meter and a QIAamp® DNA stool mini kit (Qiagen GmbH, Germany). Qualification and quantification of DNA were determined using a Nanodrop

Spectrophotometer (Thermo Fisher Scientific, Waltham, MA, USA). Extracted DNA samples were immediately stored at $-20\,°C$.

The variable region of V3-V4 of 16S rRNA genes was amplified using the forward primer Imina-V3-V4-F (5′-TCGTCGGCAGCGTCAGATGTGTATAAGAGACAG-CCTACGGGNGGC WGCAG-3′) and reverse primer Imina-V3-V4-R (5′- GTCTCGTGGGCTCGGAGATGTGTAT AAGAGACAGGACTACTACHV-GGGTATCTAATCC-3′). Cycling conditions consisted of an initial denaturation at 94 °C for 2 min, followed by 25 cycles of denaturation at 94 °C for 20 s, annealing at 57 °C for 30 s, extension at 72 °C for 30 s, and a final extension at 72 °C for 10 min. The amplified products were purified using NucleoSpin® Gel and PCR Clean-up (Macherey-Nagel Inc., USA) according to the manufacturer's protocol and sent to the Omics Sciences and Bioinformatics Center (Faculty of Science, Chulalongkorn University, Bangkok, Thailand) for further sequencing using the Illumina MiSeq platform (USA).

## Processing of 16S rRNA gene sequences and data analysis

The raw sequences were processed using bioinformatics tools. Paired-end reads were first quality trimmed using BBDUK (read quality >15 at 3′), and the primer at the 5′ end was removed using seqtk (https://github.com/lh3/seqtk). Any obtained sequence shorter than 150 bp was excluded along with its pair. Filtered pairs of sequences were corrected, merged and chimeras were removed using the DADA2 pipeline (v1.10) (*Callahan et al., 2016*). High-quality sequences were obtained at 41,843 ± 10,553 reads per sample. Taxonomy was identified using the QIIME2 classifier (v2019.1) (*Bolyen et al., 2019*) with the Greengenes version 13.8 database. Cutoff confidence was indicated at 0.7.

The PICRUSt2 pipeline package (v2.1.0_b) was used to predict the functional potential of gut microbiomes from observable 16S rRNA sequences. The KEGG pathway levels were inferred according to the instructions from the PICRUSt2 website (*Douglas & Maffei, 2020*).

## Alpha and beta diversity analysis

Alpha diversity (Chao1, Shannon) and beta diversity were calculated using the Vegan package of R software (version 2.5.6). Statistical differences in the diversity indices among the three groups were identified using the Kruskal–Wallis H test. Beta-diversity among participants was visualized *via* non-metric multidimensional scaling (NMDS) ordination based on Bray–Curtis dissimilarity matrices.

## Short-chain fatty acid analysis

Lactic acid, acetic acid, butyric acid, and propionic acid from the fecal slurries were analyzed using high-performance liquid chromatography (HPLC) (Water 1525, USA). Samples were prepared following the modified method of *Wang et al. (2019)*. The supernatant was collected using centrifugation at 13,000×g at 4 °C for 5 min and mixed with 0.2% v/v tartaric acid as the internal standard (ratio 3:1). Before injection, the mixed solution was filtered through a 0.2 μm PVDF filter (Verical, Thailand). SCFAs were isolated using an Aminex HPX-87H column (300 × 7.8 mm; Bio-Rad, Hercules, CA, SA) at 50 °C, with 8 mM sulfuric acid used as the mobile phase at a flow rate of 0.60 ml/min for 60 min and

wavelength 210 nm using a UV detector (Waters 2489). The injector was set with a split ratio of 20:1. Agilent Technologies 7890A equipment (Santa Clara, USA) was used for data collection and calculation of all parameters. Standard solutions were prepared from lactic acid, acetic acid, propionic acid and butyric acid at concentrations of 1, 2, 2.5, 5, 10 and 20 mmol/ml.

## Stool secretory IgA

Analysis of secretory immunoglobulin A (IgA) in stool samples was performed in 55 out of the 56 subjects who completed the study (19 in 3CMH, 17 in 5CMH, 19 in placebo group). Samples were collected at the baseline and after the treatment period and stored at −80 °C. All fecal samples were thawed at room temperature and weighed to 0.1 g. Phosphate-buffered saline was added to the samples, then they were mixed in a stomacher machine and centrifuged at 14,000 rpm at 4 °C for 15 min. A sample of 200 µl (10×) of supernatant was collected and diluted to 200× for the single plex human isotyping assay. IgA was measured using a Bio-Plex Pro$^{TM}$ Human Isotyping kit (cat no. 171A3101M; Bio-Rad). The Bio-Plex Pro$^{TM}$ assays were essentially immunoassays formatted on magnetic beads by captured antibodies directed against the desired biomarker coupled to the beads. The coupled beads reacted with samples containing the IgA biomarker of interest, and biotinylated detection antibody was added to create a sandwich complex. The final detection complex was formed with addition of streptavidin-phycoerythrin conjugated using the Bio-Plex® 200 system reader (Bio-Rad). The concentration of analyte bound to each bead was proportional to the median fluorescence intensity of the reporter signal.

## Statistical analysis

Statistical analyses were performed using the statistical software package SPSS version 17. Statistical significance among the three groups was determined using one way ANOVA and the Kruskal-Wallis test for parametric and non-parametric tests, respectively. Comparisons of the two groups were determined using the independent $t$-test and Mann–Whitney U-test for parametric and non-parametric tests, respectively.

Linear mixed models of bacterial relative abundance were performed for each participant using R package (2.15.0) including age, gender, carbohydrate intake, fiber intake, dose of CMH supplement and sampling period.

## RESULTS

### Demographic characteristics of study subjects

Of the 227 subjects screened, 60 were enrolled in the project at visit 1 and 56 completed the study (Fig. 1). The drop out occurred randomly due to the use of antibiotics and anti-viral treatment of subjects, which happened to occur in CMH group. Another drop-out subject was due to the non-compliance of stool collection. Therefore, the higher drop out in CMH groups was unlikely related to the testing products. The participants were randomly assigned to the three subgroups, each of which had similar demographic characteristics (Table 1). Participants in the 3CMH group were older than those in the placebo group. During the testing period, energy, and nutrient consumption values of the placebo, 3CMH and

**Table 1  Demographic characteristics of study participants at baseline.**

| Variable | Placebo | 3CMH | 5CMH | *p* value |
|---|---|---|---|---|
| Number | 20 | 19 | 17 | 0.998 |
| Age (year) | 27.7 ± 4[b] | 32 ± 5[a] | 31 ± 5[a,b] | 0.041 |
| Sex (female/male) | 15/5 | 15/4 | 13/4 | 0.958 |
| Body mass index (kg/m$^2$) | 21.6 ± 2.1 | 22 ± 2 | 21 ± 2 | 0.231 |

Notes.
All values are expressed as mean ± SD.
[a,b] Significance between sample groups by the Kruskal–Wallis test are indicated by different lowercase superscripts ($p \leq 0.05$).

**Table 2  Energy and nutrient consumption of placebo, 3CMH and 5CMH subjects during the testing period.**

| Item | Average intake/day | | | *p* value |
|---|---|---|---|---|
| | Placebo ($n = 20$) | 3CMH ($n = 19$) | 5CMH ($n = 17$) | |
| Carbohydrate (g) | 153.95 (135.98, 236.54) | 160.89 (163.54, 206.81) | 166.46 (143.71, 186.50) | 0.404 |
| Protein(g) | 59.16 (52.39, 74.36) | 61.02 (52.46, 74.32) | 61.02 (46.04, 68.38) | 0.307 |
| Fat (g) | 52.38 (42.93, 69.32) | 52.38 (44.35, 60.15) | 52.50 (37.77, 55.08) | 0.505 |
| Fiber (g) | 6.09 (4.89, 9.58) | 6.09 (5.4, 9.15) | 6.42 (4.92, 8.90) | 0.365 |
| Sugar (g) | 44.18 (40.16, 93.09) | 44.18 (48.38, 91.34) | 54.90(37.5, 69.26) | 0.396 |
| Saturated fat (g) | 16.10 (12.18, 20.56) | 16.10 (12.35, 19.27) | 15.93 (12.4, 15.61) | 0.330 |
| Energy (kcal) | 1388.28 (1182.42, 1719.00) | 1427.81 (1318.17, 1585.92) | 1482.45 (1089.36, 1448.69) | 0.341 |

Notes.
All values are expressed as median and interquartile range (IQR).

5CMH groups were calculated (Table 2). Results showed that all subjects consumed similar carbohydrate, protein, fat, fiber, and sugar contents. Therefore, the energy consumption was also similar in all groups.

## Influence of CMH on defecating conditions and gastrointestinal symptoms

Defecating conditions and gastrointestinal symptoms in the baseline period of all groups were similar (Table S4). Interestingly, stool size of participants in the 3CMH group was significantly smaller than that in the placebo group during the testing period ($p < 0.05$). The effect was still apparent even after the products were discontinued (washout period). Stool size from the 5CMH group was also smaller than that from the placebo group but did not reach statistical significance. Participants in the 3CMH and 5CMH groups showed improvement in defecating conditions including less strong smell, softened stools and reduced gastrointestinal discomfort (such as bloating, flatulence and abdominal discomfort) compared to the placebo group during the testing and washout period but without statistical significance (Table S4).

Median changes of defecating and gastrointestinal symptoms between the baseline and testing period or washout period were analyzed. Two factors including size of stool and flatulence/bloating symptoms showed significant difference between sample groups (Fig. 3). Comparing with the baseline, size of stool and flatulence/bloating symptoms of the 3CMH group reduced and reduction level was significantly lower than the placebo group ($p = 0.059$). Similarly, reduction of stool size and flatulence/bloating symptoms in

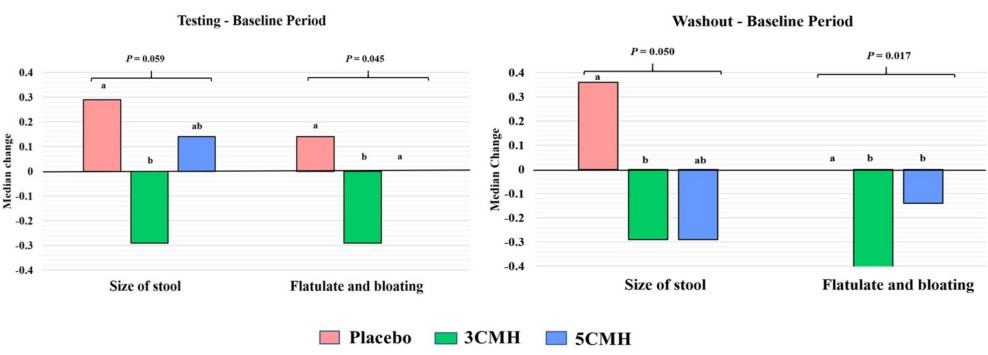

**Figure 3** **Significant change of defecating condition and gastrointestinal symptoms of placebo, 3CMH and 5CMH groups ($p \leq 0.059$).** Comparison between the three groups, different letters (a, b) show significant difference ($p \leq 0.05$).

the 3CMH group was observed in the washout period when compared to the baseline. Median level change was also significantly lower than the placebo group ($p = 0.045$). Unlike the 3CMH group, median changes of stool size and flatulence/bloating symptoms from baseline to testing period in the 5CMH group were similar to the placebo group. Flatulence and bloating symptoms were significantly greater in the 5CMH group at washout period compared with the baseline and significantly lower than the placebo group ($p \leq 0.05$).

## Influence of CMH on microbial richness and diversity

Microbial richness indices of stool samples in each group were monitored during the baseline, testing and washout periods (Fig. S1). The Chao1 index of the 3CMH group during the testing and washout periods increased compared to the baseline, while the Chao indices for the placebo and 5CMH groups decreased and were stable, respectively.

Similarly, the Shannon diversity index for the 3CMH group showed an increasing trend during the testing and washout periods compared to the baseline (Fig. S1), while Shannon diversity indices of the 5CMH and placebo groups during the baseline, testing and washout periods were stable. These results suggested that the administration of CMH at 3 g/d promoted microbial diversity and richness.

The microbial communities from all groups and time points were intermingled in non-metric multidimensional scaling plots using Bray–Curtis dissimilarity (Fig. S1). Similar to the result of alpha diversity, beta-diversity showed no differences or clusters due to CMH intake or sampling period.

## Influence of CMH on fecal microbiota

To assess specific changes in gut microbiota caused by CMH, gut microbiota from different taxa were identified from the three sampling periods (baseline, testing, and washout) (Tables S5–S7). Six microbiota phyla (*Actinobacteria, Bacteroides, Firmicutes, Lentisphaerae, Proteobacteria* and *Verrucomicrobia*) were found in all groups (Table S8). The most dominant phyla in each period were *Firmicutes, Bacteroides* and *Actinobacteria*. Dynamic changing patterns during the baseline, testing and washout periods of all phyla from the three groups were similar. No significant changes were recorded in the microbial

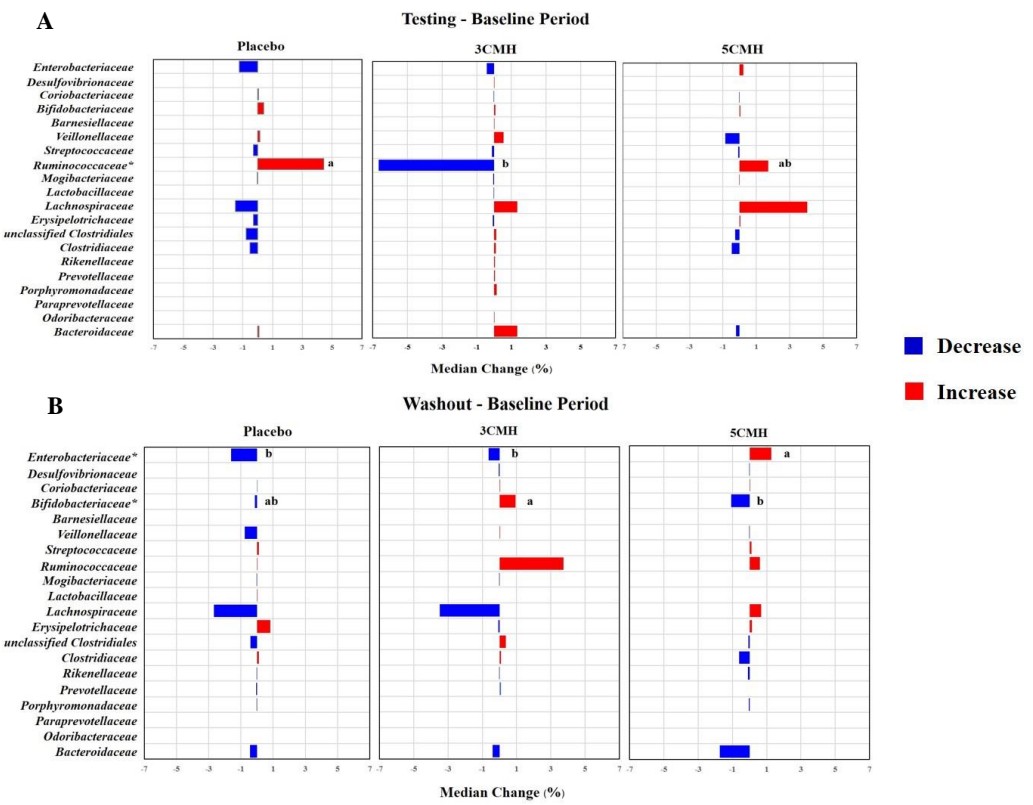

**Figure 4 Change of gut microbiota at family level for testing (A) and washout period (B) of placebo, 3CMH and 5CMH groups compared to the baseline.** For those marked with an asterisk (*), significant difference between placebo, 3CMH and 5CMH groups is shown by different letters ($p \leq 0.05$).

phyla in the CMH treatment. It was observed that *Firmicutes* and *Bacteroidetes* ratios at the testing period for both the 3CMH and 5CMH groups were lower than the placebo, although they were not significantly different (Table S8).

At the family level, changes in gut microbiota during testing period were similar in all groups (Fig. 4A). Supplementation with 3CMH promoted various microbes including *Desulfovibrionaceae, Bifidobacteriaceae, Barnesiellaceae, Veillonellaceae, Lachnospiraceae, Clostridiaceae, Rikenellaceae, Prevotellaceae, Porphyromonadaceae, Paraprevotellaceae, Odoribacteraceae, and Bacteroidaceae,* while eight families of gut microbiota in the placebo and 5CMH groups were stable (mean change = 0) and six families decreased. *Ruminococcaceae* was reduced in the 3CMH group, whereas it increased in the placebo and 5CMH groups. Median change in *Ruminococcaceae* in the placebo group was significantly greater than in the 3CMH group ($p \leq 0.05$).

After the washout period, median changes in gut microbiota were similar in all groups (Fig. 4B). However, induction of *Bifidobacteriaceae* was observed in the 3CMH group even after the products were discontinued, while showing reduction in the placebo and 5CMH groups. The amount of *Enterobacteriaceae* reduced in the 3CMH and placebo groups, with

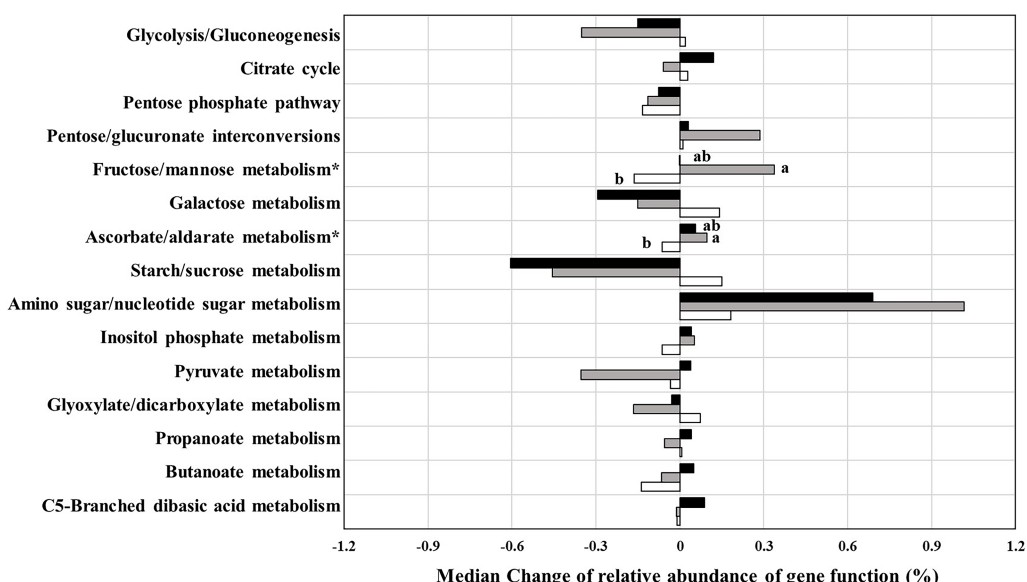

**Figure 5 Prediction of change in carbohydrate related metabolism by PICRUSt analysis from baseline to testing period between placebo (white), 3CMH (gray) and 5CMH (black) groups.** For those marked with an asterisk (*), significant difference between sample groups is shown by different letters ($p \leq 0.05$).

significantly different median change compared with the 5CMH group ($p \leq 0.05$), that showed an increase during the washout period.

To confirm the relationship of changes of three families including *Ruminococcaceae, Bifidobacteriaceae* and *Enterobacteriaceae* with CMH intake, the linear mixed effect model was performed (Table S9). Results revealed that gender, age, sampling period, carbohydrate and fiber consumption had no significant effect on changes in the three families. Supplementation with 3CMH showed a significant effect on *Ruminococcaceae* observed during testing period, while with 5CMH significantly affected *Bifidobacteriaceae* and *Enterobacteriaceae* during the washout period ($p \leq 0.05$)

At the testing period, predictions of functional analysis of gut microbiota were focused on carbohydrate-related metabolism (Fig. 5). Among the fifteen pathways, median change in relative abundance of the two pathways, ascorbate/aldarate and fructose/mannose metabolism showed significant differences between the three groups. The gene function associated with ascorbate/aldarate and fructose/mannose metabolism in the 3CMH group significantly increased more than in the placebo group ($p \leq 0.05$). However, median changes in the 5CMH group were similar to those in the placebo group, indicating that CMH intake enhanced the microbial pathways of ascorbate/aldarate and fructose/mannose metabolism.

## Influence of CMH on fecal short-chain fatty acids

The major fecal metabolite in all samples were acetic acid, followed by butyric acid, propionic acid, and lactic acid (Table 3). The SCFAs in all subjects at the baseline were similar. Intake of CMH had no effect on the four SCFAs and they were similar during

**Table 3  Short chain fatty acid concentration of feces at baseline, testing and washout.**

| SCFA | Time point | Placebo | 3CMH | 5CMH | *p*-value | | | |
|---|---|---|---|---|---|---|---|---|
| | | Median (IQR) (μmol/g feces) | Median (IQR) (μmol/g feces) | Median (IQR) (μmol/g feces) | P *vs* 3CMH *vs* 5CMH | P *vs* 3CMH | P *vs* 5CMH | 3CMH *vs* 5CMH |
| | Baseline | 17.15 (11.84, 23.31) | 21.35 (18.41, 24.24) | 21.89 (17.34, 39.97) | 0.125 | 0.193 | 0.058 | 0.347 |
| Lactic acid | Testing | 19.68 (15.02, 27.97) | 30.67 (20.21, 36.7) | 17.69 (16.44, 28.13) | 0.123 | 0.098 | 0.65 | 0.068 |
| | Washout | 17.7 (4.79, 21) | 24.36 (19.11, 31.62) | 26.47 (20.87, 29.22) | 0.015* | 0.057 | 0.005* | 0.664 |
| | Baseline | 58.21 (50.51, 79.36) | 61.42 (48.41, 79.1) | 55.4 (47.16, 66.76) | 0.588 | 0.866 | 0.377 | 0.366 |
| Acetic acid | Testing | 64.28 (45.18, 70.16) | 52.72 (44.39, 69.98) | 51.18 (44.13, 62.7) | 0.349 | 0.465 | 0.135 | 0.552 |
| | Washout | 61.78 (46.2, 75.22) | 61.78 (41.43, 77.79) | 53.83 (48.95, 71.56) | 0.947 | 0.866 | 0.927 | 0.716 |
| | Baseline | 23.72 (16.49, 27.18) (n = 19) | 23.12 (15.86, 28.83) | 24.15 (18.71, 34.85) | 0.604 | 0.942 | 0.311 | 0.476 |
| Propionic acid | Testing | 25.63 (17.58, 30.68) | 25.16 (17.11, 29.15) | 23.33 (15.82, 26.5) | 0.493 | 0.8 | 0.273 | 0.35 |
| | Washout | 24.35 (17.16, 28.04) | 24.86 (17.95, 31.01) | 24.74 (21.76, 28.49) | 0.725 | 0.536 | 0.465 | 0.862 |
| | Baseline | 29.2 (20.1, 37.1) | 29.36 (24.26, 41.67) | 29.42 (18.08, 36.87) | 0.992 | 0.955 | 0.879 | 1 |
| Butyric acid | Testing | 32.48 (24.5, 39.48) | 27.55 (22.02, 41.36) | 32.94 (24.71, 45) | 0.656 | 0.339 | 0.784 | 0.623 |
| | Washout | 35.69 (25.48, 40.45) | 26.24 (21.54, 32) | 28.34 (23.84, 38.21) | 0.2 | 0.049* | 0.537 | 0.476 |

Notes.
*Significant difference between sample groups indicated by the Kruskal–Wallis test ($p \leq 0.05$).
Descriptive statistic, interquartile range (IQR).

the testing period; however, significant differences among the three groups were observed during the washout period. Highest abundance of lactic acid was recorded in the 5CMH group followed by the 3CMH and placebo groups ($p < 0.05$). The amount of lactic acid in the 5CMH group was significantly higher than in the placebo group but similar to the 3CMH group. The level of butyric acid in the placebo group was significantly greater than in the 3CMH group.

## Influence of secretory immunoglobulin A in stools

Results revealed no significant differences in secretory IgA levels in stools among all groups both at the baseline and after the testing period (Table 4). Forty-two percent of the participants who received 3CMH treatment showed increased stool secretory IgA compared to 20% of the participants in the 5CMH and placebo groups. The increase in IgA correlated with increased stool frequency ($r = 0.303$ to $0.039$) and reduced abdominal discomfort symptoms ($r = -0.238$) during the treatment period.

## DISCUSSION

The health effects of dietary fibers have been accepted worldwide (*Carlson et al., 2018*). However, health effects of CMH have not been well defined, especially in human. To address this lack of knowledge, a randomized, double-blinded placebo-controlled design was used to determine the impact on gut health of CMH containing mannooligosaccharides. Two different dosages of CMH, 3 and 5 g/d, were investigated for gastrointestinal symptoms and defecating condition. Consumption of CMH 3 g/d improved defecating condition and

**Table 4 Quantity of Immunoglobulin A (IgA) in stool samples.**

| Stool IgA | Placebo ($n = 19$) Median (pg/mL) (IQR) | 3CMH ($n = 19$) Median (pg/mL) (IQR) | 5CMH ($n = 17$) Median (pg/mL) (IQR) | *p*-value |
|---|---|---|---|---|
| Baseline | 79069 (13216, 331439.01) | 22441.1 (7789.4, 332368.98) | 33852.5 (13573.8, 137549.8) | 0.654 |
| Testing | 50919.7 (13580.8, 332101.51) | 37519.6 (10274.7, 331891.65) | 14500.4 (3156.4, 39546.6) | 0.147 |
| Median change (IQR) | −2914.7 (−241278.22, 3065.58) | 2322.3 (−11346.5, 11236.6) | −10800.2 (−88488.5, −1495.07) | 0.190 |
| *p*-value≠ | 0.198 | 0.687 | 0.013 | |
| **Change of stool IgA** | **Placebo** ($n = 19$) | **3CMH** ($n = 19$) | **5CMH** ($n = 17$) | ***p*-value** |
| | N (%) | N (%) | N (%) | |
| Increase | 4 (21.1%) | 8 (42.1%) | 3 (17.6%) | 0.195 |
| Decrease | 8 (42.1%) | 5 (26.3%) | 12 (70.6%) | 0.027[*] |
| Same | 7 (36.8%) | 6 (31.6%) | 2 (11.8%) | 0.211 |

**Notes.**

[*]indicates significant ($p \leq 0.05$) difference.

Descriptive statistic, interquartile range (IQR).

gastrointestinal symptoms by reducing the size of stools and ameliorating flatulate/bloating symptoms (Table S4). A smaller stool is easier to expel and improves discomfort (*Bannister et al., 1987*). This effect was still observed after discontinuing CMH consumption. Prebiotics at a dose of ≤ 6 g/d reduced flatulence but a higher dose might cause intestinal bloating, pain, flatulence, or diarrhea (Gibson et al. 2010; (*Moretti et al., 2018*; *Piemontese et al., 2011*; *Scholtens, Goossens & Staiano, 2014*; *Wilson et al., 2019*). In our study, intake of 5 grams of CMH per day did not induce any flatulence/bloating or discomfort, while a dose of 3 g/d had more impact on gastrointestinal symptoms.

Prebiotics contribute to change in the gut microbial community by acting as a primary carbon source. Several reports revealed that prebiotics enhanced the intestinal microbial community (*Carlson et al., 2018*). Our results suggested that consumption of CMH at both 3 and 5 g/d did not significantly change the gut microbial richness and diversity (Fig. S1), which tended to increase after the testing period in the 3CMH group, with higher levels still maintained after the washout period.

Only a few microbiome phyla changed as the result of CMH treatments (Table S8). The *Firmicutes/Bacteroidetes* ratio of the 3CMH and 5CMH groups after the testing period tended to be lower than in the placebo group. A high *Firmicutes/Bacteroidetes* ratio has been linked with overweight and obesity (*Castaner et al., 2018*; *Verdam et al., 2013*). A positive effect of MOS on gut microbiota modulation of obesity and fat metabolism has been reported. Intake of MOS, in mice fed with a high-fat diet, decreased the *Firmicutes/Bacteroidetes* ratio (*Wang et al., 2018*). Moreover, coffee MOS also inhibited the intestinal absorption of dietary fat and enhanced fat excretion in healthy adults (*Kumao & Fujii, 2006*; *Kumao et al., 2005*).

Enhancement of *Bifidobacteriaceae, Lactobacillus, Enterobacteriaceae* and *Enterococcus* spp. was observed in CMH gut model fermentation (*Prayoonthien, Nitisinprasert & Keawsompong, 2018*; *Prayoonthien et al., 2019*). In a human trial, intake of CMH at 3 g/d enhanced human colonic *Bacteroidetes* including *Rikenellaceae, Prevotellaceae, Porphyromonadaceae, Paraprevotellaceae, Odoribacteraceae,* and *Bacteroidaceae*. These

bacteria contained hemicellulose-degrading enzymes that hydrolyzed xylans, mannans and galactomannans (*Flint et al., 2008*; *Flint et al., 2012*; *Holscher et al., 2015*). Intake of 3 g/d of CMH promoted several intestinal microbes than the 5CMH group except for *Ruminococcaceae*.

Our result was supported by PICRUSt analysis (Fig. 5). Changes in genes related to fructose/mannose metabolism in the 3CMH group increased after the testing period, with significantly higher median change than in the placebo group. Change in fructose/mannose metabolism in the 5CMH group also increased after the testing period but was lower than in the 3CMH group.

Induction of *Bifidobacteriaceae* was not observed after the testing period (Fig. 4), but was found in the 3CMH group during the washout period. This result suggested that CMH treatment indirectly introduced the conditions for enrichment of *Bifidobacteriaceae*. This trend was more evident in the washout period. Induction of *Bifidobacteriaceae* during the washout period of 5CMH was not detected, whereas the family *Enterobacteriaceae* increased. A possible explanation for this is selective utilization in small doses but loss of activity when in excess, as other microbes may become involved in fermentation (*Walton et al., 2010*).

Prebiotics are selectively fermented by the gut microbiome to produce SCFAs but effects of CMH on SCFA production remain unclear in human studies (Table 3). SCFA measurement in feces was not reflected the amount within the colon and 95% of SCFAs produced by the gut microbiome were absorbed (*Walton et al., 2010*). However, levels of lactic acid in feces of the 3CMH group were higher than in the placebo.

Prebiotics have been reported to boost the immune system through modulation of the gut microbiome (*Hemarajata & Versalovic, 2013*). The daily consumption of 3 g of CMH promoted the increase level of IgA, whereas no induction effect was observed following a high dose of CMH. Moreover, the percentage of subjects whose stool IgA value increased was higher in the 3CMH group than in the placebo and 5CMH groups. This result was consistent with other studies reporting that daily consumption of MOS at 1–3 g was more effective on the gut microbiome and immune modification than a higher dosage (*Asano et al., 2004*; *Kumao et al., 2006*; *Umemura et al., 2004a*; *Umemura et al., 2004b*; *Walton et al., 2010*).

Our experimental design showed a high rate of compliance, however there were several limitations. The lack of significant improvement in gut health could be attributed to the small sample size of the intervention study. Participants in the placebo group were mostly younger than the treatment groups. Our research samples were mostly from healthy adult females (77%). Future clinical studies are needed to provide on gender and age associated disparities in the effect of CMH on defecating conditions, gastrointestinal symptoms, and gut microbiome modulation.

## CONCLUSION

CMH is a novel potential source of prebiotics and has shown health benefits in human studies. Consumption of CMH at 3 g/d improved defecating conditions and gastrointestinal

symptoms. Intake of CMH at 3 g/d also improved the gut microbiome by increasing beneficial microbes, their metabolites and host immunity. Our results suggest that CMH has a prebiotic effect when taken at 3 g/d by healthy individuals. This finding can also increase CMH value for functional food industry application.

## ACKNOWLEDGEMENTS

The authors thank Assoc. Prof. Dr. Wanwipa Vongsangnak and Assoc. Prof. Dr. Sunee Nitisinprasert for valuable comments and discussion. The authors also acknowledge the Division of Allergy and Immunology, Department of Pediatrics, Faculty of Medicine, Chulalongkorn University for laboratory facilities and cohort study.

### Funding

This research was supported by National Research Council of Thailand (NRCT) and the postdoc program from Center for Advanced Studies for Agriculture and Food (CASAF), Kasetsart University Institute for Advanced Studies, Kasetsart University. The funders had no role in study design, data collection and analysis, decision to publish, or preparation of the manuscript.

### Grant Disclosures

The following grant information was disclosed by the authors:
National Research Council of Thailand (NRCT).
Center for Advanced Studies for Agriculture and Food (CASAF).
Kasetsart University Institute for Advanced Studies, Kasetsart University.

### Competing Interests

The authors declare there are no competing interests.

### Author Contributions

- Witida Sathitkowitchai performed the experiments, analyzed the data, prepared figures and/or tables, authored or reviewed drafts of the paper, and approved the final draft.
- Narissara Suratannon conceived and designed the experiments, performed the experiments, analyzed the data, prepared figures and/or tables, authored or reviewed drafts of the paper, and approved the final draft.
- Suttipun Keawsompong and Sunee Nitisinprasert conceived and designed the experiments, authored or reviewed drafts of the paper, and approved the final draft.
- Wanlapa Weerapakorn performed the experiments, prepared figures and/or tables, and approved the final draft.
- Preecha Patumcharoenpol analyzed the data, prepared figures and/or tables, and approved the final draft.
- Massalin Nakphaichit conceived and designed the experiments, analyzed the data, prepared figures and/or tables, authored or reviewed drafts of the paper, and approved the final draft.

## Clinical Trial Ethics

The following information was supplied relating to ethical approvals (i.e., approving body and any reference numbers):

The Institutional Review Board of the Faculty of Medicine, Chulalongkorn University, has approved the study (Ethical Application Ref: IRB No. 388/61) and registered on the Thai Clinical Trials Registry (TCTR20190426003).

## DNA Deposition

The following information was supplied regarding the deposition of DNA sequences:

The 16S rRNA gene sequence of the human gut microbiota are available at GenBank: PRJNA709129.

## Data Availability

The raw data are available in the Supplemental Files.

## Clinical Trial Registration

The following information was supplied regarding Clinical Trial registration:

TCTR20190426003

## Supplemental Information

Supplemental information for this article can be found online at http://dx.doi.org/10.7717/peerj.12158#supplemental-information.

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
