# Peer review of "A randomized trial to evaluate the impact of copra meal hydrolysate on gastrointestinal symptoms and gut microbiome"

_PeerJ, doi:10.7717/peerj.12158_

## Round 0.1 · original submission · Minor Revisions

Both reviewers have noted several issues which need to be addressed. In particular, the authors should indicate the limitations of their study, and address some of the questions around statistical analyses, particularly those raised by reviewer 2.

The authors should also check their manuscript to ensure the quality of English is acceptable.

Reviewer 1 ·

Basic reporting

no comment

Experimental design

no comment

Validity of the findings

no comment

Additional comments

The manuscript describes a clinical trial to evaluate the impact of CMH (two doses) on gut health in healthy adults. The major limitations are that the sample size is quite small and the age of subjects was significant different in 3gCMH group compared to others. Several of the key finding was found with 3gCMH intake. It is therefore extremely important to have an appropriate statistical model to analyze the data. In addition, the manuscript is quite lengthy and the data presentation and language need to be improved. Please add Faecal calprotectin as marker for gut immune status.
Introduction
1) Line 48-49, please update the definition of prebiotics “The current scientific definition of a prebiotic was developed by a panel of experts in microbiology, nutrition, and clinical research convened by ISAPP in 2016.
2) For line 51-52, please revise. Prebiotics are frequently equated with dietary fibers, not include many others. SCFA is just one of the mechanism.
Materials and method
1) Line 82, please describe the MOS using degree of polymerization.
2) The description of the study design (line 94-100) is confusing, please simplify.
3) Please provide more details and reference for stool sample preparation.
4) Suggest to use more appropriate statistical model, such linear mixed effects models to compare among three groups (line 193-194)
Results
1) Table 2: Does the carbohydrate exclude sugar? Please clarify. Although the P value is not significant (0.062), 3CMH and 5CHM seems to have higher carb. Please discuss how this could impact your results.
2) Move Table 3 into supplement and show statistical analysis in Figure 3.
3) Figure 2, please discuss the higher drop out in CMH groups.
4) Please move Figure 4 into supplement and add the p value. Please als provide beta diversity analysis.
5) Please use Table instead of Figure to present data in Figure 5.
6) Missing Figure legend to point out the representative symbol for each group.

Discussion: Please 1) simplify the discussion, and don’t describing result repeatedly; 2) discuss why many effects were observed in 3g CMH group but not 5g CMH groups; 3) provide explanation why many effect on gut microbiota was observed during wash out , but not intervention.
1) This statement is not accurate (line 293): there is age difference.
2) Line 318: at family level

Reviewer 2 ·

Basic reporting

Given two dosage of daily consumption were tested, it would be helpful to provide a brief background and rationale to test two different dosages in the introduction (and elaborate more in discussion section).

Tables 2 and 3 – were means ± SD presented? Please clarify in a footnote. Figures 6 and 7, the meaning of superscripts “a” and “b” next to some bars should be described in the footnote. This was not made clear in Table 1 either.

Why figure 3 presented only the significant differences in flatulate/bloating and size of stool? What about the changes of the other 6 indicators on defecating conditions and gastrointestinal symptoms? Would it be fair to conclude that 3g CMH improved defecating conditions if only two out of 8 indicators showed a significant improvement?

Figures 4 and 5 did not indicate statistical significance at all. Was the reduction in the Firmicutes/Bacteroidetes ratio statistically significant? The information seems to be missing in the results section. P-values should be consistently reported.

A discussion of potential limitations is missing. As no study is perfect, acknowledging limitations is important. For example, could self-reported stool chart and dietary record be free of measurement errors? Were all participants consume the distributed supplementation as indicated? Could differences in fiber or carbohydrate intake not influence the results? Could the results be generalizable to larger population with different health status and other demographic characteristics?

The sample was predominantly female, given the biological difference, it may be helpful to comment on potential gender differences.

Experimental design

More details on subject selection should be provided, including recruiting methods, location, and procedure etc.

It would be helpful to comment on the validity of stool size and characteristics instruments (i.e., stool record chart), their clinical implications, and potential reporting bias etc.

It would be helpful to provide the details of sample size calculation, the main outcome measure(s), the effect size, alpha level, power etc. to allow replication of the calculation.

An important advantage of Kruskal-Wallis test and Mann-Whitney test is that they do not assume that the data are normally distributed. However, these non-parametric approaches, which substitute ranks for the original values, could be less powerful than their parametric equivalents. Moreover, practically ANOVA was not very sensitive to deviations from normality. Given that many of the outcomes were continuous measures, the blanket application of non-parametric tests to ordinal and interval/ratio outcomes may have sacrificed efficiency.

A few baseline characteristics appear to be unequally allocated, particularly, the placebo group was significantly younger compared with the treatment groups, and 5CMH group seem to have lower median body weight. With only 5 baseline characteristics measures included in Table 1, it is difficult to judge whether baseline characteristics were well balanced and how well randomization performed. Since BMI is a function of height and weight, Table 1 essentially just showed mean levels of 3 variables -- age, sex, and BMI -- by study groups.

Related to this, average consumption of dietary fiber was over 70% higher among 3CMH group compared with that among placebo group (9.27 vs 5.44 g). Although the omnibus Kruskal-Wallis test showed a p value slightly larger than 0.05, the difference is likely significant with Mann-Whitney test. Similarly, 3CMH group had 23% higher consumption of carbohydrate. Albeit at lower concentration, many foods have prebiotic effects, including garlic, onion, wheat, honey, banana, barley, tomato, soybean, cow’s milk, peas, beans, etc. These differences warrant a careful evaluation and discussion, given that unbalanced dietary patterns could be a strong alternative explanation to treatment effect.

Validity of the findings

The abstract should include a few important qualifications related to the sample characteristics, i.e., the number of participants, predominantly female, aged 18-40, normal BMI, and healthy status etc. I would also suggest toning down the general conclusion, especially the statement in the abstract that 3g CMH “could be considered a source for a future prebiotic product”, given the limited generalizability of the present study to a more diverse population (with a wide variety of age, health conditions, physical activity levels, dietary patterns etc.).

Additional comments

The manuscript assessed the potential prebiotic effect of copra meal hydrolysate. The authors reported that daily 3g/d CMH supplementation over a period of 21 days was beneficial to gut health. In general, the manuscript is well written, the analysis is rigorous, and the topic is of scientific interest.

---

## Round 0.2 · Minor Revisions

The authors have responded appropriately to Reviewer 2's comments, as noted by that reviewer. They have also largely responded to those comments raised by Reviewer 1 (who was unavailable to re-review).

However there are still some issues that require clarification or further revision, before the manuscript could be considered acceptable for publication.

Responses to Reviewer 1's comments:
For Tables 2 and 3, the authors note they have used median +/- SD. Is that correct? It is unusual to use this combination, I would expect either mean and SD, OR median and interquartile range. Please confirm.

In the limitations that have been added in response to Reviewer 2's comment, the authors need to note that the proposed future studies (lines 395-396) would enable the results to be more generalizable.

Further comments

Line 30 (and elsewhere): The authors refer to reduced risk of obesity, but I think making any reference to such a long-term effect based on the relatively short-term intervention presented here should be done with caution. Particularly given (as noted in the next point) the authors base this on a change in the Firmicutes / Bacteroidetes ratio. They do not provide sufficiently strong evidence that this ratio has been altered, therefore basing discussion on potential effects of a change in this ratio is very speculative.

Line 279: It is stated that "Firmicutes and Bacteroidetes ratios for the testing period for both the 3CMH and 5CMH groups were lower than the placebo (Table S8)" however although this is apparently so, the p-values are not significant, therefore it cannot be said that these values were lower. At most, it would be reasonable to state that they were apparently lower, but this was not significant. Because of this lack of significance, it is not possible to draw the conclusion that these ratios were sensitive to CMH.

First paragraph of the discussion:
I don't believe describing the potential functional food value of CMH is the right way to start the discussion. It should be about the science. I suggest re-wording this paragraph as follows:

The health effects of dietary fibers have been accepted worldwide (Carlson et. al. 2018). However, health effects of CMH have not been well defined, especially in humans. To address this lack of knowledge, a randomized double-blind placebo-controlled design was used to determine the impact on gut health of CMH containing mannooligosaccharides.

The possible relevance for functional foods could be included in the conclusions.

While the use of English has improved, there are still numerous typographical and grammatical errors. For example:
Line 22 (and elsewhere, for example line 36): The parentheses used around "CMH" are of a different font size and possible a different font style. This should be corrected throughout.
Line 28: "With," - the comma is not required.
Line 32: There should be a comma, rather than a full stop, after Bacteroidetes.
Line 45: Jarret et al ref, 2019 the font size is incorrect (also line 52, Topping reference)
Line 72: "No official dose recommendation for prebiotics in have been reported." This sentence does not make sense. Do you mean "in the human diet"? Please clarify.
Line 184: "were examined" should be "were identified"
Line 394: "Our research samples were mostly healthy adult females" - do you mean your research participants were mostly adult females? Please correct.

These are just a few examples of the errors present. The authors should please carefully check the revised manuscript for such errors, and correct them.

Reviewer 2 ·

Basic reporting

The authors have addressed my comments sufficiently. I have no further comments.

Experimental design

N/A

Validity of the findings

N/A

---

## Round 0.3 · accepted · Accept

I believe the authors have addressed the previously raised concerns, and the manuscript is acceptable for publication.